# The Role of Gene Expression Dysregulation in the Pathogenesis of Mucopolysaccharidosis: A Comparative Analysis of Shared and Specific Molecular Markers in Neuronopathic and Non-Neuronopathic Types of the Disease

**DOI:** 10.3390/ijms252413447

**Published:** 2024-12-15

**Authors:** Karolina Wiśniewska, Magdalena Żabińska, Aneta Szulc, Lidia Gaffke, Grzegorz Węgrzyn, Karolina Pierzynowska

**Affiliations:** Department of Molecular Biology, Faculty of Biology, University of Gdansk, Wita Stwosza 59, 80-308 Gdansk, Poland; karolina.wisniewska@phdstud.ug.edu.pl (K.W.); magdalena.zabinska@phdstud.ug.edu.pl (M.Ż.); aneta.szulc@phdstud.ug.edu.pl (A.S.); lidia.gaffke@ug.edu.pl (L.G.)

**Keywords:** mucopolysaccharidosis, neurodegeneration, transcriptomics

## Abstract

Mucopolysaccharidosis (MPS) comprises a group of inherited metabolic diseases. Each MPS type is caused by a deficiency in the activity of one kind of enzymes involved in glycosaminoglycan (GAG) degradation, resulting from the presence of pathogenic variant(s) of the corresponding gene. All types/subtypes of MPS, which are classified on the basis of all kinds of defective enzymes and accumulated GAG(s), are severe diseases. However, neuronopathy only occurs in some MPS types/subtypes (specifically severe forms of MPS I and MPS II, all subtypes of MPS III, and MPS VII), while in others, the symptoms related to central nervous system dysfunctions are either mild or absent. The early diagnosis of neuronopathy is important for the proper treatment and/or management of the disease; however, there are no specific markers that could be easily used for this in a clinical practice. Therefore, in this work, a comparative analysis of shared and specific gene expression alterations in neuronopathic and non-neuronopathic MPS types was performed using cultures of cells derived from patients. Using transcriptomic analyses (based on the RNA-seq method, confirmed by measuring the levels of a selected gene product), we identified genes (including *PFN1*, *ADAMTSL1*, and *ABHD5*) with dysregulated expression that are common for all, or almost all, types of MPS, suggesting their roles in MPS pathogenesis. Moreover, a distinct set of genes (including *ARL6IP6* and *PDIA3*) exhibited expression changes only in neuronopathic MPS types/subtypes, but not in non-neuronopathic ones, suggesting their possible applications as biomarkers for neurodegeneration in MPS. These findings provide new insights into both the molecular mechanisms of MPS pathogenesis and the development of differentiation method(s) between neuronopathic and non-neuronopathic courses of the disease.

## 1. Introduction

Although fibroblasts might not initially appear to be an obvious model for studying neurodegeneration, they share several significant characteristics with neurons. Both cell types originate from the ectoderm, suggesting common molecular mechanisms, including signaling pathways and gene regulatory processes. This feature is critical for elucidating the pathogenesis of neurodegenerative diseases. Additionally, fibroblasts exhibit disruptions in cellular pathways, such as oxidative stress, autophagy, and metabolism, which are also essential factors in neurodegeneration. Fibroblasts can reflect systemic cellular changes that influence neuronal function, especially in genetic disorders where the consequences of mutations remain unclear. Furthermore, fibroblasts are found near blood vessels, in the meninges, and within the choroid plexus of the brain and spinal cord, where they play critical roles in maintaining central nervous system (CNS) function. Importantly, fibroblasts are easily accessible through minimally invasive procedures, enabling regular patient monitoring, and their stable cultures facilitate long-term studies, making them a valuable model for research into neurodegenerative mechanisms [1,2].

One of the aforementioned genetic disorders is mucopolysaccharidosis (MPS). This group of metabolic diseases arises from the disruption (reduction or complete absence) of lysosomal hydrolase activity, leading to the accumulation of glycosaminoglycans (GAGs). This accumulation damages cells, tissues, and the entire organism. Based on the defective enzyme and the type of stored GAG(s), 12 to 14 types and subtypes of MPS are currently recognized, depending on the criteria adopted [3]. The characteristics of MPS types/subtypes are presented in Table 1.

Although MPS represents a heterogeneous group of diseases, several symptoms are typical across all subtypes [4,5]. Special attention should be given to symptoms related to CNS which occur in some, but not all, MPS types/subtypes, as indicated in Table 1 (see Refs. [6,7,8]). Patients with neuronopathic forms of MPS usually require round-the-clock care [9]. Although patients with non-neuronopathic MPS types may experience some neurological problems, they are not dominant in clinical settings [5,10,11,12].

The availability of a specific therapy largely depends on the type of MPS [13]. One of the treatment methods is enzyme replacement therapy (ERT), which involves supplying the missing enzyme to the body. A similar approach is used in hematopoietic stem cell transplantation (HSCT) for young patients (up to 2 years old). The transplanted bone marrow or blood cells (from either umbilical or peripheral blood) from healthy individuals migrate to the patient’s organs and tissues, synthesizing properly functioning enzymes [14,15,16,17]. ERT and HSCT are registered for MPS types I, II, IVA, VI, and VII [4,5,18]. For type III MPS, where somatic symptoms are relatively mild but CNS disorders are especially severe, there is currently no registered treatment method [19,20].

Despite the current understanding of the etiology of MPS, available treatment methods can only alleviate the symptoms of the disease, and they do so to a limited extent [5,21]. The effectiveness of therapies largely depends on the patient’s age (the older the patient, the more advanced the changes that cannot be reversed). Additionally, enzymes and stem cells significantly struggle to reach poorly vascularized tissues, such as bone tissue and heart valves, as well as the brain, which is protected by the blood–brain barrier. As a result, these tissues hardly respond to treatment [4,5,7,22,23,24,25]. On the other hand, the early diagnosis of MPS is essential for achieving the best therapeutic outcomes. Identifying whether the disease has a neurodegenerative component or not would be especially beneficial, as it would guide the choice of treatment and provide better insights into the patient’s prognosis [5].

Research on gene expression regulation in genetic diseases, such as MPS, is crucial for understanding the pathogenesis of these disorders. Such studies allow for the identification of changes in gene activity that lead to abnormal cellular functions and the development of the disease. This kind of research also enables the development of new therapeutic strategies that could focus on correcting abnormal gene expression, which cannot be restored by ERT or HSCT. Moreover, understanding the mechanisms of gene expression regulation facilitates the advancement of molecular diagnostics, which can lead to the earlier detection of genetic diseases, the prompt initiation of treatment, and the effective monitoring of its course. In the case of MPS, it may also help with predicting the neuronopathic component. As a result, such research helps to improve the quality of life for patients and increase the effectiveness of therapies [26,27,28].

Since neuronopathic and non-neuronopathic types of MPS differ in symptoms, it is likely that these differences arise not only from the kind of accumulated GAG(s), but also from other molecular processes. In fact, the significant modulation of MPS pathomechanism has been demonstrated to be related to the dysregulation of expression of specific genes [21,29]. Therefore, the aim of this study was a comparative analysis of shared and specific gene expression alterations in neuronopathic and non-neuronopathic MPS types using fibroblasts derived from patients. This research should help us to identify potential disease markers and their specific symptoms, with a particular focus on neurodegeneration.

## 2. Results

### 2.1. Indentification of Transcripts with Changed Expression Levels in More than One MPS Type/Subtype

In the first stage of this study, we aimed to identify transcripts whose expression is disrupted more than once among the 11 tested MPS types/subtypes. The analysis of the number of genes with altered expression levels in MPS cells compared to control cells indicated approximately 300 such transcripts for MPS II, IVA, and VI, and almost 900 for MPS IIIA, IIIB, IVB, and IX. Furthermore, for each MPS type/subtype, expression disruptions in at least half of these genes were also observed in at least one other type/subtype of MPS (Figure 1). In general, the disruption of the expression of a large number of transcripts was observed not in an individual, but rather in multiple types/subtypes of MPS, both neuronopathic and non-neuronopathic, indicating common pathways in the pathogenesis of the MPS disease.

A more detailed analysis involved determining the number of transcripts with altered expression levels depending on the number of MPS types/subtypes. It revealed, for example, 399 transcripts with altered expression levels compared to the control cells in at least 2 different MPS types/subtypes, 85 transcripts with altered expression levels in at least 6 MPS types/subtypes, and 8 transcripts with altered expression levels in at least 10 MPS types/subtypes. It also revealed one transcript with altered expression levels in all 11 tested MPS types/subtypes compared to the control (Figure 2).

### 2.2. Transcripts with Altered Levels in Most of MPS Types/Subtypes

The list of transcripts with altered expression levels in at least 10 MPS types/subtypes, for which the log_2_ fold change value exceeded 2.5 or −2.5 (log_2_FC > 2.5 or <−2.5), is shown as a heatmap in Figure 3 and Table 2, with exact log_2_FC values. This list includes transcripts, such as the up-regulated *PFN1*, encoding the protein profilin, which showed increased expression in all MPS types/subtypes. Up-regulation was also observed for genes like *ADAMTSL1* (coding for ADAMTS Like 1 or Punctin-1, a putative secretory protein), *MFAP5* (coding for Microfibril-Associated Protein 5), *SH3BP5* (coding for SH3 Domain-Binding Protein 5), and *CAPG* (coding for Gelsolin-Like Capping Actin Protein). The analysis also identified down-regulated genes, including *C1D* (coding for C1D Nuclear Receptor Corepressor), *ABHD5* (coding for Abhydrolase Domain Containing 5, Lysophosphatidic Acid Acyltransferase), *LY6K* (coding for Lymphocyte Antigen 6 Family Member K), and *PLCB4* (coding for Phospholipase C Beta 4).

### 2.3. PFN1—A Gene Whose Expression Is Enhanced in All MPS Types/Subtypes

Since *PFN1* was the only gene showing increased expression levels in all MPS types/subtypes, the immunodetection of the encoded protein, profilin, was performed in MPS cells. The profilin levels were significantly increased in all MPS types, specifically from about three-fold to about seven-fold, depending on the MPS type/subtype (Figure 4). This also confirmed the reliability of the transcriptomic results, obtained by an independent method.

### 2.4. Genes with Efficiency of Expression Changed Specifically Only in Neuronopathic MPS Types/Subtypes

The aim of the next stage of this study was to identify genetic factors that could potentially be responsible for the neuronopathic component of the described disease. An analysis was therefore conducted to isolate genes whose expression changes are specific only to the neuronopathic MPS types/subtypes (I, II, IIIA, IIIB, IIIC, IIID, and VII). Genes whose expression changes were also observed in non-neuronopathic MPS types (IVA, IVB, VI, and IX) were excluded from this analysis. The results demonstrated that among hundreds of genes with altered expression levels in each of the neuronopathic MPS types/subtypes, between 30 and 50% of the transcripts with changed expression levels also appeared in at least one other neuronopathic MPS type/subtype. In total, between 140 and approximately 400 genes (depending on the type/subtype of the disease) were found to possibly be associated with nervous system disorders (exactly 418 unique genes whose expression is significantly altered in neuronopathic MPS types/subtypes but not in non-neuronopathic types) (Figure 5).

The analysis of the number of transcripts with altered expression levels depending on the number of neuronopathic MPS types/subtypes showed values ranging from 322 (for at least two types/subtypes) to 1 (for six types/subtypes). There was no single gene whose expression was altered in all seven neuronopathic MPS types (Figure 6).

The list of genes whose expression was altered compared to control cells in at least five neuronopathic MPS types/subtypes, without changes in expression levels in non-neuronopathic MPS types/subtypes, is shown in Figure 7 and Table 3. A significant decrease in the expression of the *ARL6IP6* gene, which encodes ADP Ribosylation Factor Like GTPase 6 Interacting Protein 6, was observed in six out of seven MPS neuronopathic types/subtypes (no significant changes were observed in type IIID). Other genes that exhibited altered expression levels compared to control cells in five neuronopathic MPS types/subtypes included down-regulated *C11orf58* (corresponding to Chromosome 11 Open Reading Frame 58) and *MINOS1* (coding for Myosin X), and up-regulated *RPN2* (coding for Ribophorin II), *PDIA3* (coding for Protein Disulfide Isomerase Family A Member 3), and *VASN* (coding for Vasorin).

An analysis was also conducted to determine the number of transcripts whose expression was changed (either up- or down-regulated) based on the log_2_ fold change (log_2_FC) in expression relative to the control cells (Appendix A). The list of transcripts showing a three-fold increase/decrease in MPS cells compared to control cells is presented as a heatmap in Appendix A, along with the exact fold change values. High fold change values (log_2_FC > 3 or <−3) in the expression were frequently observed in individual MPS types/subtypes. Notable down-regulated genes in MPS compared to controls included *RARRES2* (coding for Retinoic Acid Receptor Responder 2) (MPS IIIA and IIIB) and *CXCL8* (coding for C-X-C Motif Chemokine Ligand 8) (MPS I and IIIA), while up-regulated genes compared to controls included *MFGE8* (coding for Protein Containing Milk Fat Globule EGF and Factor V/VIII Domain) (MPS I and IIIA) and *RPLP2* (coding for Ribosomal Protein Lateral Stalk Subunit P2) (MPS I and IIIB).

## 3. Discussion

MPS is a group of lysosomal storage diseases associated with impaired GAG degradation, leading to their accumulation in various tissues. Different types/subtypes of MPS exhibit significant clinical variability—some patients have severe neurodegenerative symptoms, while others mainly experience peripheral organ involvement (Table 1).

Currently, it is not possible to predict whether, and if yes—to what extent, a patient will develop neurological symptoms, as the neurodegenerative component can occur with varying severity even within the same type/subtype of MPS [30]. The ability to predict such a component would be of significant importance for patients and their families. First, it would allow for the early implementation of therapies targeting the protection of neurological functions, such as intrathecal or intracerebroventricular ERT or gene therapy delivered intrathecally to target the brain directly. Second, it would enable better planning of specialized and rehabilitative care, which could improve the quality of life and slow down the progression of symptoms. Early recognition of the risk of neurodegeneration would also help with adjusting expectations and preparing caregivers for the specific challenges associated with progressing neurological changes, influencing a comprehensive approach to long-term care. Unfortunately, despite attempts to find specific markers for neuronopathy in MPS, only complex bioinformatic analyses could be offered to date, which, despite their utility in theoretical analyses, are hardly applicable in clinical practice [31]. Therefore, further research is needed to better understand which factors may influence the development of neurodegenerative symptoms in MPS.

Since finding minimally invasive markers of neurodegeneration is of significant scientific and medical importance, the aim of this study was to conduct a comparative analysis of shared and specific gene expression alterations in neuronopathic and non-neuronopathic types of MPS using fibroblasts derived from patients. Such studies enabled (i) the selection of transcripts/genes whose expression is altered in multiple MPS types/subtypes, allowing for the determination of their involvement in common disease mechanisms, and (ii) the selection of genes whose expression is altered in comparison to control cells only in neuronopathic types/subtypes, which could be considered potential markers of neurological disorders.

The use of fibroblasts derived from patients in neurodegeneration research may seem surprising, but it is highly valuable. Their collection is less invasive compared to obtaining nerve tissue cells. Furthermore, as mentioned earlier, fibroblasts and neurons originate from the same embryonic germ layer, which means they can reflect some key molecular changes typical of neurodegeneration in MPS. Recent studies also indicated that fibroblasts are present around blood vessels, in the meninges, and in the choroid plexus of the brain and spinal cord, where they play important roles for CNS. Fibroblasts in the meninges support the immune response by secreting cytokines, and perivascular fibroblasts play a crucial role in scar formation after nerve inflammation. Limiting their proliferation can reduce scarring and improve motor functions [1,2]. Hence, fibroblasts can serve as a practical and more accessible alternative model to nerve cells for studies on pathogenic mechanisms and potential therapeutic targets in neurodegenerative diseases [32]. Such studies for rare metabolic diseases with a neurodegenerative component are commonly conducted using fibroblast models [33,34,35,36,37].

Transcriptomic analysis conducted with fibroblasts derived from patients with neuronopathic and non-neuronopathic types of MPS (Table 2) revealed that expression alterations of many genes (relative to controls) occur in various MPS types. Some of these gene expression alterations are shared among all/multiple types/subtypes of the disease, while others are specific only to neuronopathic types/subtypes, making them potential markers for neurodegeneration.

The analyses conducted in this work showed that at least half of the genes with altered expression levels compared to control cells in a given MPS type also appear in other types/subtypes of the disease (Figure 1 and Figure 2). The list of genes whose expression is altered in at least 10 types/subtypes of MPS, with a log_2_ fold change (log_2_FC) greater than 2.5 or less than −2.5, includes up-regulated genes *PFN1, ADAMTSL1, MFAP5, SH3BP5*, and *CAPG*, as well as down-regulated genes *C1D, ABHD5, LY6K*, and *PLCB4* (Figure 3 and Figure 4; Table 3).

The *PFN1* gene is the only gene that exhibits increased expression levels in the cells of all MPS types/subtypes studied in this work when compared to control cells. It plays a crucial role in the development of the skeletal system and the maintenance of bone homeostasis. Previous studies demonstrated that disturbances in the levels of profilin-1, the protein encoded by *PFN1*, have been detected in individuals with skeletal disorders, such as dwarfism, facial deformities, and altered bone structure and size. These findings highlighted the importance of PFN1 function in bone health and its potential connection to the pathophysiology of MPS [38]. A heterozygous deletion of the *PFN1* gene has been detected in patients with Paget’s disease of bone, a chronic progressive bone disorder of late onset. Paget’s disease of bone is characterized by the abnormal activation of osteoclasts, which leads to bone pain, deformities, and fractures [39]. Heterozygous deletions of the PFN1 gene have also been observed in patients with a severe phenotype of Miller–Dieker syndrome, a rare genetic disorder caused by a contiguous gene deletion. This syndrome is characterized by lissencephaly type 1 (a condition resulting from impaired neuronal migration), facial dysmorphism, seizures, and severe intellectual disability. These findings suggest that profilin-1 may play a critical role not only in the proper development of the skeletal system but also in the nervous system, supporting its involvement in both bone formation and neurogenesis [40]. In zebrafish models with *PFN1* mutations, motor deficits are observed at early life stages, associated with changes in the structure of motor neurons. It has also been shown that mutations in *PFN1* lead to disruptions in actin dynamics, resulting in protein aggregation and the degeneration of motor neurons in *Drosophila* models. Studies on mice with PFN1 mutations demonstrate that alterations in this protein function cause symptoms resembling amyotrophic lateral sclerosis in humans. These mice exhibit progressive muscle weakness, coordination problems, and the degeneration of motor neurons in the spinal cord. Histological analyses also revealed disruptions in the cytoskeleton of neurons, indicating damage to the mechanisms stabilizing actin. Furthermore, certain mutated forms of profilin-1 exhibit prion-like properties and can induce the transformation of natural TDP43 into toxic conformational structures upon entering cells with unmutated prifilin-1 [41,42,43,44,45]. It would be consistent with recent data on the pathogenesis of MPS III, where the accumulation of protein aggregates such as TDP43, APP, beta-amyloid, and alpha-synuclein was observed in cells obtained from patients [46]. Dysregulation of or mutations in the *PFN1* gene have also been implicated in Fragile X syndrome, spinal muscular atrophy, Huntington’s disease, Parkinson’s disease, and adrenoleukodystrophy [47].

The *ADAMTSL1* gene, encoding ADAMTS Like 1 or Punctin-1, also appears to be of interest in terms of developmental disorders. Mutations in this gene have also been observed in Microcephaly Facial, Dysmorphism Ocular, and Multiple Congenital Anomaly Syndrome. This disease is characterized by symptoms similar to MPS, particularly in terms of facial dysmorphia (square face with a prominent jaw, wide and flat nasal bridge, short philtrum, and protruding ears). Additionally, symptoms of this syndrome include congenital glaucoma, myopia, retinal detachment, hearing loss, dental abnormalities, kidney anomalies, brain vessel malformations, hypothyroidism, and distal limb deformities [48].

Mutations in the *ABHD5* gene, encoding Abhydrolase Domain Containing 5, Lysophosphatidic Acid Acyltransferase, have been identified in patients with Chanarin–Dorfman Syndrome, a rare autosomal recessive disorder characterized by triglyceride accumulation in various tissues. Patients with this syndrome display a range of symptoms that overlap with those seen in MPS, such as hepatomegaly, liver dysfunction, muscle weakness, growth delay, cataracts, hearing loss, and intellectual disability [49].

On the basis of above facts, the association of genes identified in this work, which show altered expression levels in MPS cells with the clinical features observed in patients with both MPS and other mentioned diseases, appears indisputable. However, at the current stage of our knowledge, it is difficult to deduce whether a disturbance in the expression levels of a particular gene, or perhaps an entire group of genes, is responsible for the appearance of characteristic symptoms.

In the next phase of the study, an analysis was conducted to identify genes with altered expression levels exclusively in neuronopathic types/subtypes of MPS (MPS I, II, IIIA, IIIB, IIIC, IIID, and VII) compared to control cells, which were not altered in non-neuronopathic types/subtypes. This analysis revealed that between 30 and 50% of the transcripts with changed expression levels were also present in at least one other neuropathic type/subtype of MPS (Figure 5 and Figure 6). In total, 418 genes were identified whose expression was significantly altered in the neuronopathic MPS types/subtypes but not in non-neuronopathic types, suggesting their potential involvement in CNS disorders. The list of genes with altered expression levels in at least five neuropathic MPS types/subtypes includes up-regulated *RPN2*, *PDIA3*, and *VASN*, as well as down-regulated *ARL6IP6*, *C11orf58*, and *MINOS1* (Figure 7; Table 3).

The *ARL6IP6* gene is involved in key functions related to the regulation of alternative splicing, which is crucial for the proper functioning of the nervous system. Disruptions in its function can lead to abnormal neuronal differentiation and disturbances in mitochondrial and endoplasmic reticulum homeostasis. Additionally, the delivery of a correct copy of this gene via gene therapy has been shown to reduce neuroinflammation and neurodegeneration, as demonstrated in in vivo and in vitro models of hereditary spastic paraplegia [50]. Moreover, the product of this gene has been found to play a role in regulating the activity of BACE1, the beta-site amyloid precursor protein (APP) cleaving enzyme-1, which is involved in the processing of APP into beta-amyloid [51]. The reduction in its levels, as demonstrated in this study, could therefore explain the increase in beta-amyloid and its precursor levels, as well as their tendency to aggregate, which has been observed in previous research on MPS [46,52,53].

Protein disulfide-isomerase A3, encoded by the *PDIA3* gene, also plays a significant role in protein proteostasis by facilitating proper protein folding. A product of mutated *PDIA3* is prone to aggregate formation, which aberrantly interacts with ER chaperones. This disrupts the biogenesis and signaling of integrins, essential adhesive molecules that support synaptic activity [54]. An interesting piece of research on the role of *PDIA3* in nervous system damage has been conducted using a mouse model of traumatic brain injury. Findings from studies on wild-type mice (*PDIA3*^+/+^) and *PDIA3* knockout mice (*PDIA3*^−/−^) revealed that the absence of *PDIA3* significantly improved cognitive function and reduced the contusion volume caused by trauma. Additionally, *PDIA3* deletion was associated with decreased apoptosis, mitigated neuroinflammation, and reduced oxidative stress, highlighting its potential role in modulating these processes during brain injury [55]. Similar conclusions were reached by researchers who demonstrated an age-dependent increase in PDIA3 levels in the amygdala, entorhinal cortex, and ventral hippocampus of Alzheimer’s disease model mice (3×Tg-AD) [56]. In contrast, healthy mice exhibited an age-dependent decrease in PDIA3 levels. Moreover, immunohistochemical analysis revealed a direct correlation between cellular levels of beta-amyloid and PDIA3 across all examined brain regions, suggesting a potential interaction in the pathogenesis of Alzheimer’s disease [56].

The most important limitation of this study was using only one cell line per MPS type/subtype. However, MPS is a very rare disease with poor availability of biological materials. For example, there are only single fibroblast lines of MPS IIIC and MPS IX, and only two commercially available MPS IIID fibroblast lines (https://www.coriell.org/1/NIGMS/Additional-Resources/Available-Products#cell; last accessed on 30 November 2024). On the other hand, one should note that, in most cases, the results were similar in groups of all neuronopathic and all non-neuronopathic types of MPS in the cell lines tested, which indicates their reliability. Moreover, each experiment was repeated several times (three for Western blotting and four for transcriptomics). Indeed, such a limitation arising for the ultra-rarity of some genetic diseases, including MPS, has been explained previously, indicating that the results based on analyses of single cell line can be reliable under certain conditions, especially when using a suitable number of repeats of experiments and obtaining data showing similar tendency of changes in similar kinds of diseases [21,29,31,46]. One might also suggest that more cell lines should be investigated in the cases of MPS types where such lines are available. However, comparing transcriptomic results of experiments where some types are represented by several cell lines while others are represented by single cell lines would be incoherent and statistical analyses would be unreliable. Therefore, we decided to stay with a single cell line per each MPS type/subtype.

One might also ask whether factors unrelated to MPS type/subtype could influence the results of our transcriptomic analysis. Indeed, patients being donors of fibroblasts used in this study differed in sex, race, and age of the biological material withdrawal, as indicated in Table 4. However, we were not able to find any correlations between these parameters and transcriptomic results. Thus, in light of the identified correlations between the expression of certain genes and MPS neuronopathy (discussed above), we conclude that it is likely that the observed effects are related to neuronopathic and non-neuronopathic types/subtypes of MPS rather than other factors, like the sex, race, or age of the patient.

In summary, our study identified a set of genes with dysregulated expression common across all types of MPS, such as *PFN1*, *ADAMTSL1*, and *ABHD5*, emphasizing their role in the disease’s pathogenesis. Additionally, a distinct set of genes, including *ARL6IP6* and *PDIA3*, exhibited expression changes specific to neuronopathic MPS types/subtypes, suggesting their possible applications as biomarkers for neurodegeneration in MPS. These findings provide new insights into the molecular mechanisms underlying MPS pathogenesis and differentiation between neuronopathic and non-neuronopathic courses of the disease, the latter being of special significance in MPS types where patients may represent one of two such courses, like MPS I and MPS II. They also highlight opportunities for developing more precise diagnostic and therapeutic strategies.

## 4. Materials and Methods

### 4.1. Cell Lines and Cell Cultures

Cell lines (skin fibroblasts) of MPS patients were purchased commercially (Coriell Institute for Medical Research, Camden, NJ, USA) (Table 4). The HDFa cells were used as controls. Cultures of cells were incubated in the DMEM medium with 10% Fetal Bovine Serum and a standard mixture of antibiotics. A temperature of 37 °C, humidity of 95%, and CO_2_ saturation of 5% were kept throughout the experiments.

### 4.2. Transcriptomic Analyses

#### 4.2.1. RNA Isolation and Purification

To isolate and purify total RNA, 5 × 10^5^ cells were passaged and incubated in the medium (Section 4.1) overnight. Cell lysis was conducted via homogenization with a QIAshredder column in the presence of guanidinium isothiocyanate and β-mercaptoethanol. An RNeasy Mini Kit (Qiagen, Hilden, Germany) and Turbo DNase (Life Technologies, Life Technologies, Carls- bad, CA, USA) were used to extract RNA molecules. Nano Chip RNA and the Agilent 2100 Bioanalyzer system (Agilent Technologies, Santa Clara, CA, USA) were employed to assess the RNA sample quality. Samples from 4 independent biological repeats (i.e., 4 independent cell cultures with independent RNA isolation procedures) were used in further analyses.

#### 4.2.2. Transcriptomic (RNA-Seq) Analysis

An Illumina TruSeq Stranded mRNA Library Prep Kit was used for obtaining mRNA libraries. Following reverse transcription, the HiSeq4000 system (Illumina, San Diego, CA, USA) was used to sequence cDNA libraries. After sequencing the parameters of the reactions, we used FastQC version v0.11.7 to determine the following values: 150 bp paired-end, 40 million raw reads, and 12 Gb of the raw data/sample. The mapping of raw readings was performed to the GRCh38 human reference genome (Ensembl; Hisat2 v. 2.1.0 software). The Cuffquant and Cuffmerge software (v. 2.2.1) and the GTF Homo_sapiens.GRCh38.94.gtf file (Ensembl database; https://www.ensembl.org/index.html, as of 19 February 2019) were used to calculate the levels of transcripts. To normalize the expression values by means of the FPKM algorithm, Cuffmerge software was employed with the library-norm-method classic-fpkm parameter. The BioMart interface for the Ensembl gene database (https://www.ensembl.org/info/data/biomart/index.html, as of 19 February 2019) was used for the annotation and classification of transcripts. Raw RNA-seq data were deposited in the NCBI Sequence Read Archive (SRA) database (accession no. PRJNA562649).

#### 4.2.3. Statistical Analysis

R v3.4.3 software was used to perform statistical analysis. One-way analysis of variance (ANOVA) on log_2_(1 + x) values was employed for data which had a continuous normal distribution. Student’s post hoc t-test with Bonferroni correction was employed to analyze statistical significance between two groups. The Benjamini–Hochberg method was used to calculate the false discovery rate (FDR). The Ensembl database (BioMart interface) (https://www.ensembl.org/info/data/biomart/index.html, as of 19 February 2019) was used for transcript classification.

### 4.3. Analysis of Protein Levels via Western Blotting

Fibroblasts from cell cultures (6 × 10^5^ cells; Section 4.1) were lysed using the buffer consisting of 1% Triton X-100, 0.5 mM EDTA, 150 mM NaCl, and 50 mM Tris (pH 7.5). A mixture of protease and phosphatase inhibitors (Roche Applied Science, Penzberg, Germany; #05892791001 and #11873580001) was employed to protect proteins from degradation. Following centrifugation for 10 min, at 12,000 rpm and temperature of 4 °C, the lysates were used for the analysis of protein levels. The WES system (WES—Automated Western Blots with Simple Western; ProteinSimple, San Jose, CA, USA) was used for automatic Western blotting. This system employs the capillary electrophoresis-based separation of proteins and immunological detection direction in each capillary (without using traditional SDS-PAGE and transfer to the membrane). The 12–230 kDa separation module (#SM-W003) was used for separating proteins. The profilin-1 protein was detected with the Anti-Mouse (#DM-002) or Anti-Rabbit (#DM-001) detection module and anti-profilin-1 antibody (#PA5-17444, Thermo Fisher Scientific, Waltham, MA, USA). As a loading control, the Total Protein Module (#DM-TP01, ProteinSimple, San Jose, CA, USA) was used.

The capillary electrophoresis-based automatic Western blotting system (the WES system) allows for the use of highly standardized procedure, resulting in the high reproducibility of results and the more precise quantitative determination of levels of investigated proteins than in traditional Western blotting. Moreover, the method is significantly shorter (3–4 h from loading samples to obtaining the final results) than traditional Western blotting and requires significantly lower amounts of studied proteins and antibodies [57,58,59,60,61,62,63,64,65,66].

## Figures and Tables

**Figure 1 ijms-25-13447-f001:**
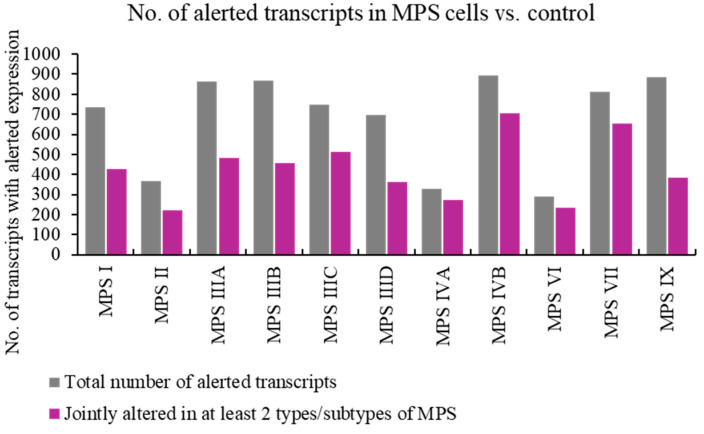
Total number of transcripts with altered levels of expression (at FDR < 0.1; *p* < 0.1) in cells of different MPS types/subtypes relative to control cells, and those concerning jointly neuronopathic and non-neuronopathic types.

**Figure 2 ijms-25-13447-f002:**
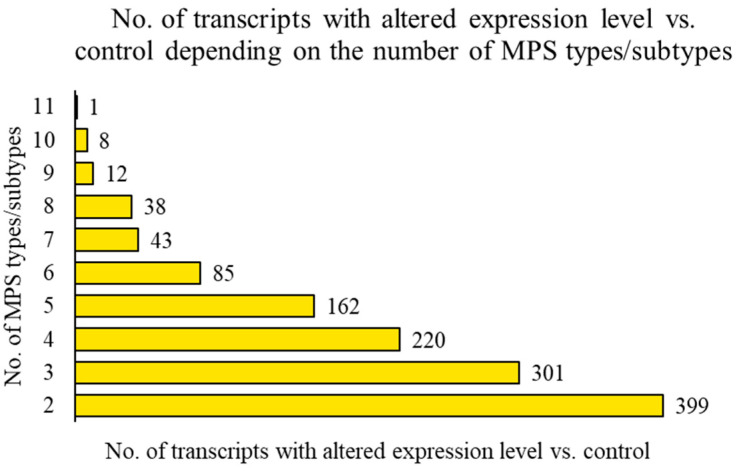
Number of transcripts with altered levels of expression (at FDR < 0.1; *p* < 0.1) in cells of different MPS types/subtypes relative to control cells in relation to the number of MPS types where such differences occur (1 transcript in 11 MPS types, 8 transcripts in 10 MPS types, and so on).

**Figure 3 ijms-25-13447-f003:**
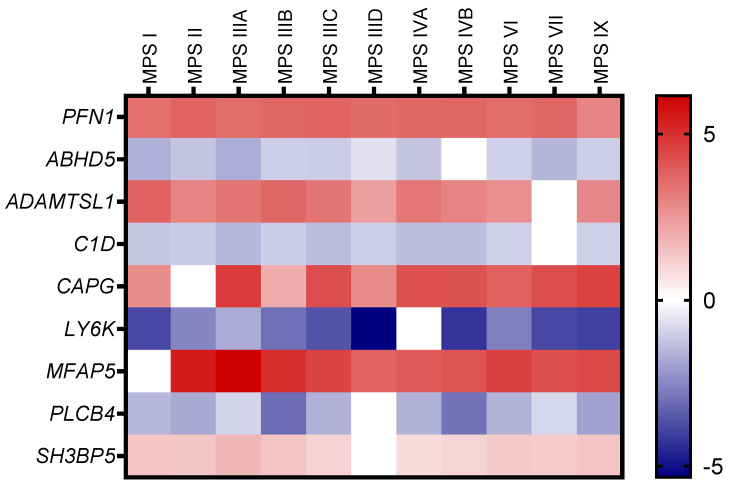
A heatmap presentation (created with HeatMapper software v. 2.8) of genes with altered expression levels in at least 10 MPS types/subtypes, for which the log_2_ fold change value exceeded 2.5 or −2.5 (log_2_FC > 2.5 or <−2.5).

**Figure 4 ijms-25-13447-f004:**
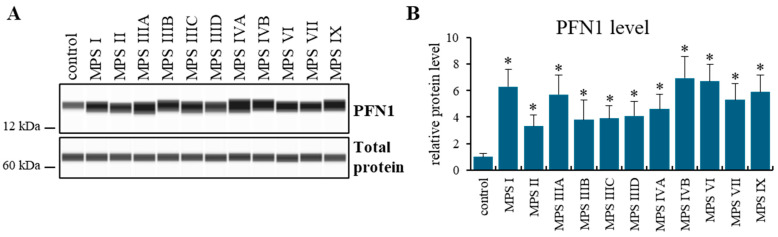
Levels of profilin-1 (PFN1 protein, the *PFN1* gene product) in control cells and in fibroblast derived from all tested MPS types/subtypes, as assessed by automatic Western blotting (the WES system, based on capillary electrophoresis and immunoblotting conducted inside each capillary). Representative blots (**A**) (the picture prepared using a piece of software which is an integrated part of the WES—Automated Western Blots with Simple Western; ProteinSimple, San Jose, CA, USA) and (**B**) (quantification of results, i.e., mean values from three independent biological experiments with error bars representing SD) are demonstrated. In panel (**A**), the Total Protein Module (#DM-TP01, Protein Simple, San Jose, CA, USA) was used to determine the loading control. Statistically significant differences (in two-way ANOVA) relative to the control (at *p* < 0.05) are indicated in panel (**B**) by asterisks.

**Figure 5 ijms-25-13447-f005:**
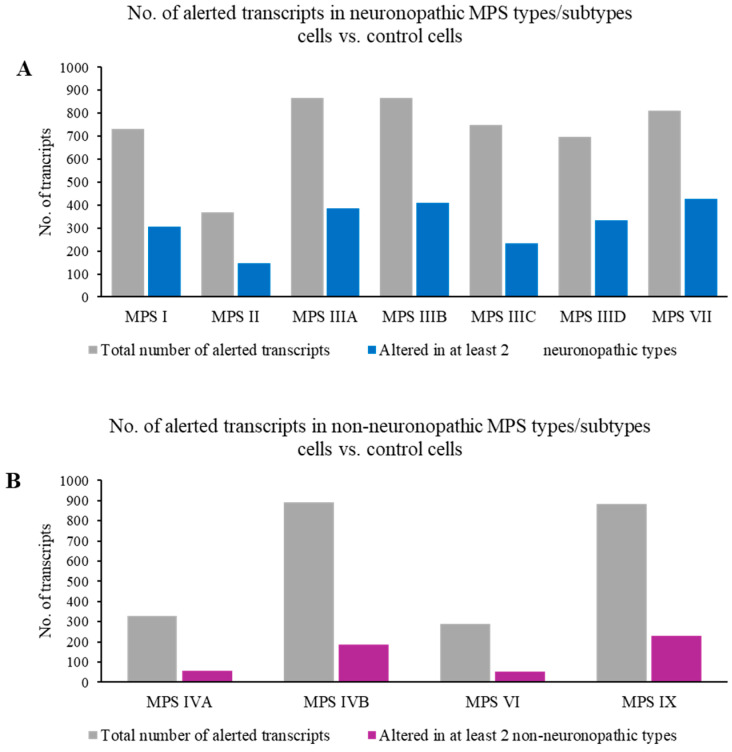
Number of transcripts with altered levels of expression (at FDR < 0.1; *p* < 0.1) in cells of different neuronopathic (**A**) and non-neuronopathic (**B**) MPS types/subtypes relative to control cells, with an indication of the number of specific transcripts altered in at least two neuronopathic (**A**) or non-neuronopathic (**B**) MPS types/subtypes.

**Figure 6 ijms-25-13447-f006:**
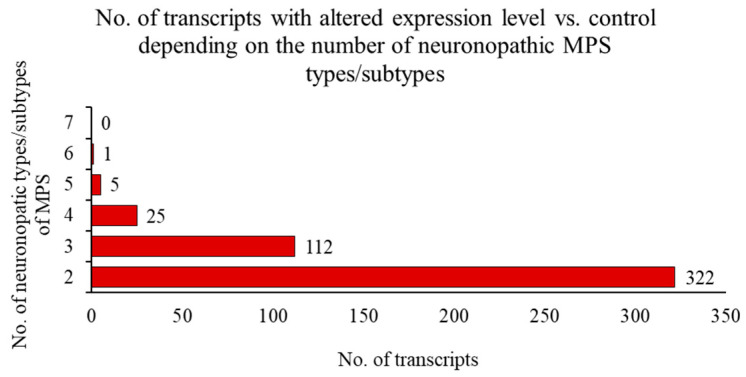
Number of transcripts with altered levels of expression (at FDR < 0.1; *p* < 0.1) in cells of different neuronopathic MPS types/subtypes relative to control cells in relation to the number of neuronopathic MPS types/subtypes where such differences occur (no (0) transcripts in 7 neuronopathic MPS types, 1 transcript in 6 neuronopathic MPS types, 5 transcripts in 5 neuronopathic MPS types, and so on).

**Figure 7 ijms-25-13447-f007:**
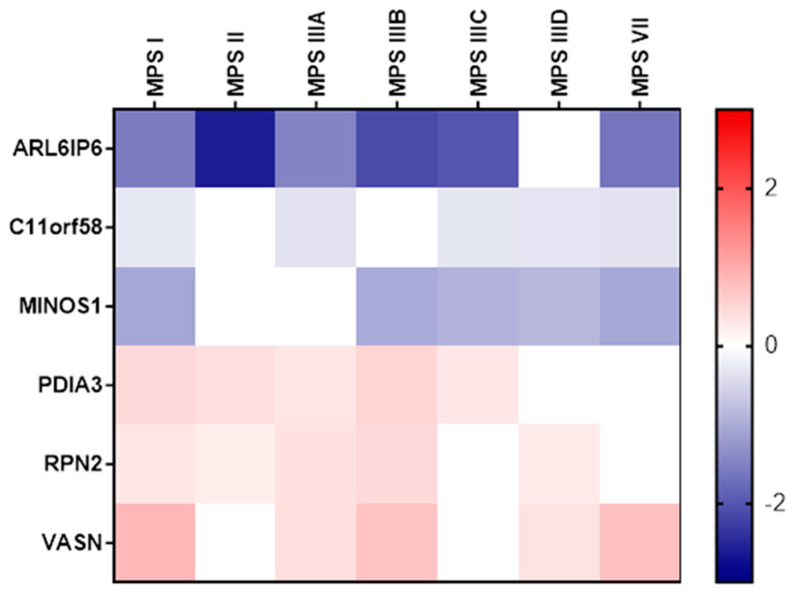
A heatmap presentation (created with HeatMapper software v. 2.8) of genes whose expression was altered compared to control cells in at least 5 neuronopathic MPS types/subtypes, without changes in expression in non-neuronopathic MPS types/subtypes.

**Table 1 ijms-25-13447-t001:** Characteristics of classical/conventional types of MPS [4,5].

MPS Type	Defective Gene	Deficient Enzyme	Stored GAG(s) ^a^	Neurological Symptoms
MPS I	*IDUA*	α-L-iduronidase	HS, DS	Impaired cognitive function, language, speech abilities, behavioural abnormalities (excessive silencing), sleeping problems, and/or epileptic seizures
MPS II	*IDS*	Iduronidase-2-sulfatase	HS, DS	Developmental delay, mental retardation, and behaviour problems (aggression and over-excitability)
MPS IIIA	*SGSH*	Heparan-N-sulfatase	HS	Developmental delay, cognitive impairment, behavioural disorders (impulsivity, aggression, anxiety disorders, and autistic behaviour), and sleeping problems
MPS IIIB	*NAGLU*	α-Nacetylglucosaminidase		
MPS IIIC	*HGSNAT*	Heparan α-Glucosaminide N-acetyltransferase		
MPS IIID	*GNS*	N-Acetylglucosamine-6-sulfatase		
MPS IVA	*GLANS*	N-Acetylglucosamine- 6-sulfate sulfatase	C6S, KS	Absence or mild neurological disorders as a consequence of secondary disturbances
MPS IVB	*GLB1*	b-Galactosidase	KS	
MPS VI	*ARSB*	N-acetylglucosamine4-sulfatase (arylsulfatase B)	DS, C4S	-
MPS VII	*GUSB*	β-Glucuronidase	HS, DS, C4S, C6S	Impaired cognitive, language, and speech abilities; behavioural abnormalities; sleep problems; and/or epileptic seizures
MPS IX	*HYAL1*	Hyaluronidase	Hyaluronan	-
MPS X	*ARSK*	Arylsulfatase K	DS	-

^a^ HS, heparan sulfate; DS, dermatan sulfate; KS, keratan sulfate; C6S, chondroitin 6-sulfate; C4S, chondroitin 4-sulfate; -, no or few neurological symptoms.

**Table 2 ijms-25-13447-t002:** Transcripts with altered expression levels in at least 10 types/subtypes of MPS compared to control cells, with log_2_FC > 2.5 or <−2.5 (X, changes smaller than log_2_FC > 2.5 or <−2.5).

MPS Type	Changes in Levels (in log_2_FC) in Specific Transcripts (MPS vs. Control)								
	*PFN1*	*ABHD5*	*ADAMTSL1*	*C1D*	*CAPG*	*LY6K*	*MFAP5*	*PLCB4*	*SH3BP5*
MPS I	3.4	−1.7	3.8	−1.2	2.7	−3.8	X	−1.5	1.4
MPS II	3.7	−1.2	3.0	−1.1	X	−2.6	5.5	−1.8	1.3
MPS IIIA	3.5	−1.7	3.3	−1.5	4.7	−1.8	6.1	−0.9	1.8
MPS IIIB	3.6	−1.0	3.6	−1.1	2.0	−3.0	5.0	−3.1	1.4
MPS IIIC	3.7	−1.1	3.3	−1.4	4.3	−3.6	4.5	−1.7	1.1
MPS IIID	3.5	−0.7	2.3	−1.1	2.8	−5.3	3.8	X	X
MPS IVA	3.6	−1.2	3.3	−1.4	4.2	X	3.9	−1.7	0.9
MPS IVB	3.7	X	3.0	−1.4	4.1	−4.3	4.1	−3.0	1.0
MPS VI	3.5	−1.0	2.7	−1.0	3.8	−2.7	4.6	−1.6	1.3
MPS VII	3.7	−1.5	X	X	4.3	−3.8	4.2	−0.9	1.2
MPS IX	2.9	−1.1	2.9	−1.0	4.6	−4.0	4.4	−2.0	1.4

**Table 3 ijms-25-13447-t003:** Transcripts with altered expression levels in at least 5 neuronopathic types/subtypes of MPS compared to control cells, without concurrent expression changes in non-neuropathic MPS types/subtypes.

Transkcript	Number of Transcripts with Common Expression Changes in at Least 5 Neuronopathic MPS Types vs. Control Cells	Up (↑) or Down (↓) Regulation	Neuronopathic MPS Types
*ARL6IP6*	6	↓	I, II, IIIA, IIIB, IIIC, VII
*C11orf58*	5	↓	I, IIIA, IIIC, IIID, VII
*RPN2*	5	↑	I, II, IIIA, IIIB, IIID
*PDIA3*	5	↑	I, II, IIIA, IIIB, IIIC
*VASN*	5	↑	I, IIIA, IIIB, IIID, VII
*MINOS1*	5	↓	I, IIIB, IIIC, IIID, VII

**Table 4 ijms-25-13447-t004:** Characteristics of cell lines used in the study.

MPS Type	Stored GAG(s) *	Defective Enzyme	Mutation(s)	Catalog Number of the Cell Line **/Patient’s Characteristics: Sex ***/Race ****/Age (in Years) at the Time of Sample Collection
MPS I	HS, DS	α-L-iduronidase	p.Trp402Ter/ p.Trp402Ter	GM00798/F/C/1
MPS II		2-iduronate sulfatase	p.His70ProfsTer29	GM13203/M/C-H/3
MPS IIIA	HS	N-sulfoglucosamine sulfhydrolase	p.Glu447Lys/ p.Arg245His	GM00879/F/C/3
MPS IIIB		α-N-acetylglucosaminidase	p.Arg626Ter/ p.Arg626Ter	GM00156/M/C/7
MPS IIIC		Acetyl-CoA:α-glycosaminide acetyltransferas	p.Gly262Arg/ pArg509Asp	GM05157/M/U/8
MPS IIID		N-acetylglucosamine 6-sulfatase	p.Arg355Ter/ p.Arg355Ter	GM05093/M/A-I/7
MPS IVA	KS, CS	N-acetylglucosamine- 6-sulfate sulfatase	p.Arg386Cys/ p.Phe285Ter	GM00593/F/C-M/7
MPS IVB		β-galactosidase	p.Trp273Leu/ p.Trp509Cys	GM03251/F/C/4
MPS VI	DS, C4S	N-acetylglucosamine- 4-sulfatase (arylsulfatase B)	Not determined	GM03722/F/B/3
MPS VII	HS, DS, CS	N-acetylgalactosamine 4-sulfatase	p.Trp627Cys/ p.Arg356Ter	GM00121/M/A-A/3
MPS IX	HA	Hyaluronidase	p.Glu268Lys/ c.37bp-del;14bp-ins at nt 1361	GM17494/F/U/14
Control line (HDFa)	None	N/A	N/A	N/A

* Abbreviations: HS, heparan sulfate; DS, dermatan sulfate; KS, keratan sulfate; C6S, chondroitin 6-sulfate; C4S, chondroitin 4-sulfate. ** According to the Coriell Institute; N/A, not applicable. *** Abbreviations: F, female; M, male; U, unknown. **** Abbreviations: A-A, African-American; A-I, Asian-Indian; B, Black; C, Caucasian; C-H, Caucasian-Haitian; C-M, Caucasian-Mexican; U, unknown.

## Data Availability

Raw data are available in the NCBI Sequence Read Archive (SRA), under accession no. PRJNA562649.

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
