# Peer review of "The Role of Gene Expression Dysregulation in the Pathogenesis of Mucopolysaccharidosis: A Comparative Analysis of Shared and Specific Molecular Markers in Neuronopathic and Non-Neuronopathic Types of the Disease"

_ijms, 2024, doi:10.3390/ijms252413447_

Round 1
Reviewer 1 Report
Comments and Suggestions for Authors
The research and possible identification of disease markers (MPS), particularly those related to neurodegeneration, is interesting and brings innovation to what was previously known. This is a very interesting and well-done study, although some changes are suggested that could make the article easier to read. Basically, the Introduction, Results and Discussion should be shortened.
Major criticism
As the authors state, a major limitation of this study was the use of only one cell line per MPS type/subtype. Although MPS are rare diseases, the availability of cultured fibroblasts of the different types, which are kept frozen in cell banks in several European countries, is possible and not difficult to obtain. The use of other cell lines (different from those commercially available) would probably provide new data that would enrich this study. The analysis of transcriptomic results from single cell lines may increase the knowledge of the different expression of clinical symptoms of these complex pathologies.
Other Comments
-The Introduction is well written and the literature on the topic is well reviewed; however, it could be shortened since there are several sentences that could be eliminated due to repetitive concepts or because they are already explicit in Table I.
- The description of the results should be clearly shortened.
-The first paragraph of the Discussion, although well written, is repetitive and re-explains "Table 1. Characteristics of classical/conventional types of MPS", only re-quoting Table 1 would seem sufficient.
Author Response
The research and possible identification of disease markers (MPS), particularly those related to neurodegeneration, is interesting and brings innovation to what was previously known. This is a very interesting and well-done study, although some changes are suggested that could make the article easier to read. Basically, the Introduction, Results and Discussion should be shortened.
Reviewer’s comment 1:
Major criticism
As the authors state, a major limitation of this study was the use of only one cell line per MPS type/subtype. Although MPS are rare diseases, the availability of cultured fibroblasts of the different types, which are kept frozen in cell banks in several European countries, is possible and not difficult to obtain. The use of other cell lines (different from those commercially available) would probably provide new data that would enrich this study. The analysis of transcriptomic results from single cell lines may increase the knowledge of the different expression of clinical symptoms of these complex pathologies.
Response 1:
Indeed, we have used one cell line per MPS type/subtype, and we know that this is a limitation of the study. However, MPS is a very rare disease with poor availability of biological materials, especially for some MPS types/subtypes. For example, there are only single fibroblast lines of MPS IIIC and MPS IX, and only two MPS IIID fibroblast lines commercially available (https://www.coriell.org/1/NIGMS/Additional-Resources/Available-Products#cell; last accessed on 30 November, 2024). On the other hand, please note that in most cases the results were similar in groups of all neuronopathic and all non-neuronopathic types of MPS in the cell lines tested, which indicates their reliability. Moreover, each experiment was repeated several times (3 for Western-blotting and 4 for transcriptomics), as indicated in the manuscript (lines 167 and 430, respectively). Indeed, this limitation has been explained previously, indicating that the results are reliable despite using one cell line per each MPS type [Gaffke et al. Int J Mol Sci 2020; 21: 1204. Rintz et al. Int J Mol Sci 2020; 21: 3194. Gaffke L. et al., Biochem Biophys Res Commun. 2023; 665: 107-117. Wisniewska et al. Biochem Biophys Res Commun 2024; 733: 150718]. We would also like to indicate that one of advantage of this study is using cells derived from patients with all types/subtypes of MPS. Using several cells lines from each type/subtype would make necessary (from technical and financial reasons) to focus on one or a few types of MPS which, however, would make it impossible to answer the main question asked in this paper. Furthermore, comparing transcriptomic results of experiments where some types are represented by several cell lines while others are represented by single cell lines would be incoherent, and statistical analyses unreliable. Therefore, we decided to stay will a single cell line per each MPS type/subtype. We have included a detailed description of this limitation and arguments indicating the reliability of the results of the study into the manuscript (lines 388-405 in the revised manuscript).
Other Comments
Reviewer’s comment 2:
-The Introduction is well written and the literature on the topic is well reviewed; however, it could be shortened since there are several sentences that could be eliminated due to repetitive concepts or because they are already explicit in Table I.
Response 2:
Introduction has been shortened, as recommended by the editor, especially by deleting the text describing symptoms already mentioned in Table 1.
Reviewer’s comment 3:
The description of the results should be clearly shortened.
Response 3:
The description of results was modified (also according to comments of Reviewer 3) and shortened somewhat. However, we did not shorten this chapter drastically, as in our opinion this would cause a loss of clarity of presentation.
Reviewer’s comment 4:
The first paragraph of the Discussion, although well written, is repetitive and re-explains "Table 1. Characteristics of classical/conventional types of MPS", only re-quoting Table 1 would seem sufficient.
Response 4:
As recommended by the Reviewer, the first paragraph of Discussion has been shortened significantly. The revised paragraphs consisted of only two sentences with quoting to Table 1.
Reviewer 2 Report
Comments and Suggestions for Authors
Authors in this manuscript investigated the Role of Gene Expression Dysregulation in the Pathogenesis of Mucopolysaccharidosis. Overall, the manuscript is well written. I have some comments that need to be addressed are given below:
1. Autor should add relevant reference in table 1.
2. This The first stage of this study aimed to identify transcripts whose expression is
disrupted more than once among the 11 tested MPS types/subtypes.
The underlined English mistake in the above sentence should be corrected.
3. In figure A what is the total protein? Is it Actin or GAPDH? Or is it any other loading control? This should be mentioned clearly.
Author Response
Authors in this manuscript investigated the Role of Gene Expression Dysregulation in the Pathogenesis of Mucopolysaccharidosis. Overall, the manuscript is well written. I have some comments that need to be addressed are given below:
Reviewer’s comment 1:
Autor should add relevant reference in table 1.
Response 1:
References were added to Table 1.
Reviewer’s comment 2:
This The first stage of this study aimed to identify transcripts whose expression is
disrupted more than once among the 11 tested MPS types/subtypes.
The underlined English mistake in the above sentence should be corrected.
Response 2:
This sentence has been corrected (lines 108-109).
Reviewer’s comment 3:
In figure A what is the total protein? Is it Actin or GAPDH? Or is it any other loading control? This should be mentioned clearly.
Response 3:
The Total Protein Module (#DM-TP01, Protein Simple, San Jose, CA, USA) was used to determine the loading control. This module measures the combined concentrations of all proteins present in the investigated sample, being considered the optimal loading control. This information is provided in the legend to Figure 4 (lines 167-168).
Reviewer 3 Report
Comments and Suggestions for Authors
This paper provides an in-depth analysis of the role of dysregulated gene expression in the pathology of MPS, with particular reference to the comparative analysis of neuronal and non-neuronal types of MPS. The study is well-designed and provides new insights into understanding the molecular mechanisms of MPS. However, there are some issues that require modifications or reasonable explanations from the authors:
1. Have the authors considered the possibility that differences in gene expression between different MPS types/subtypes may be influenced by other factors, such as patient age, gender, or disease severity?
2. The study used only fibroblasts as subjects, which may not fully reflect changes in nerve cells; what was the basis for the authors' choice?
3. For gene expression changes specific to neuronal-type MPS, such as ARL6IP6 and PDIA3, have the authors explored potential links between these gene changes and neurodegenerative symptoms?
4. The article is well structured and logical, but certain sections may need to be further simplified to improve readability. For example, could the results section be further divided into subsections to allow readers to better follow the different findings?
5. In the discussion section, can the authors provide more information about how the findings compare to the existing literature and how these findings affect our understanding of MPS?
Author Response
This paper provides an in-depth analysis of the role of dysregulated gene expression in the pathology of MPS, with particular reference to the comparative analysis of neuronal and non-neuronal types of MPS. The study is well-designed and provides new insights into understanding the molecular mechanisms of MPS. However, there are some issues that require modifications or reasonable explanations from the authors:
Reviewer’s comment 1:
- Have the authors considered the possibility that differences in gene expression between different MPS types/subtypes may be influenced by other factors, such as patient age, gender, or disease severity?
Response 1:
We thank the reviewer for this comment. We have addressed this possibility, and described the analysis in the revised manuscript (lines 406-414). However, we were not able to find any correlations between these parameters and transcriptomic results. Thus, in the light of the identified correlations between expression of certain genes and MPS neuronopathy (discussed above), we conclude that it is likely that the observed effects are related to neuronopathic and non-neuronopathic types/subtypes of MPS rather than other factors, like sex, race or age of the patient.
Reviewer’s comment 2:
- The study used only fibroblasts as subjects, which may not fully reflect changes in nerve cells; what was the basis for the authors' choice?
Response 2:
The rationale of the use of fibroblasts in studies on neuronopathic and non-neuronopathic types/subtypes of MPS is explained in Introduction. See lines 35-48 for details.
Reviewer’s comment 3:
- For gene expression changes specific to neuronal-type MPS, such as ARL6IP6 and PDIA3, have the authors explored potential links between these gene changes and neurodegenerative symptoms?
Response 3:
The potential links between expression of ARL6IP6 and PDIA3 genes (and levels/activities of their products) and neurodegeneration are discussed in the manuscript. See lines 358-369 for ARL6IP6, and lines 370-387 for PDIA3.
Reviewer’s comment 4:
- The article is well structured and logical, but certain sections may need to be further simplified to improve readability. For example, could the results section be further divided into subsections to allow readers to better follow the different findings?
Response 4:
As recommended by the Reviewer, the Result section has been divided into four subsections (2.1, 2.2., 2.3, and 2.4.).
Reviewer’s comment 5:
- In the discussion section, can the authors provide more information about how the findings compare to the existing literature and how these findings affect our understanding of MPS?
Response 5:
The indication of the impact of this study on our understanding of MPS is provided in the Discussion (lines 415-424).
Round 2
Reviewer 2 Report
Comments and Suggestions for Authors
All my comments have been addressed.